# A bio-functional polymer that prevents retinal scarring through modulation of NRF2 signalling pathway

Bhav Harshad Parikh[1,2,13], Zengping Liu[1,2,3,13], Paul Blakeley[2], Qianyu Lin[4], Malay Singh[1,5], Jun Yi Ong[1], Kim Han Ho[1], Joel Weijia Lai[6], Hanumakumar Bogireddi[2], Kim Chi Tran[2], Jason Y. C. Lim[4,7], Kun Xue[4], Abdurrahmaan Al-Mubaarak[2], Binxia Yang[1], Sowmiya R[1], Kakkad Regha[1,2], Daniel Soo Lin Wong[2], Queenie Shu Woon Tan[1], Zhongxing Zhang[4], Anand D. Jeyasekharan[8], Veluchamy Amutha Barathi[2,3,9], Weimiao Yu[1,5], Kang Hao Cheong[6], Timothy A. Blenkinsop[10], Walter Hunziker[1,11], Gopal Lingam[2,3,12], Xian Jun Loh[4,7✉] & Xinyi Su[1,2,3,12✉]

One common cause of vision loss after retinal detachment surgery is the formation of proliferative and contractile fibrocellular membranes. This aberrant wound healing process is mediated by epithelial-mesenchymal transition (EMT) and hyper-proliferation of retinal pigment epithelial (RPE) cells. Current treatment relies primarily on surgical removal of these membranes. Here, we demonstrate that a bio-functional polymer by itself is able to prevent retinal scarring in an experimental rabbit model of proliferative vitreoretinopathy. This is mediated primarily via clathrin-dependent internalisation of polymeric micelles, downstream suppression of canonical EMT transcription factors, reduction of RPE cell hyper-proliferation and migration. Nuclear factor erythroid 2–related factor 2 signalling pathway was identified in a genome-wide transcriptomic profiling as a key sensor and effector. This study highlights the potential of using synthetic bio-functional polymer to modulate RPE cellular behaviour and offers a potential therapy for retinal scarring prevention.

[1] Institute of Molecular and Cell Biology (IMCB), Agency for Science, Technology and Research (A*STAR), Singapore, Singapore. [2] Department of Ophthalmology, Yong Loo Lin School of Medicine, National University of Singapore, Singapore, Singapore. [3] Singapore Eye Research Institute (SERI), Singapore, Singapore. [4] Institute of Materials Research and Engineering (IMRE), Agency for Science, Technology and Research (A*STAR), Singapore, Singapore. [5] Bioinformatics Institute (BII), Agency for Science, Technology and Research (A*STAR), Singapore, Singapore. [6] Science, Mathematics and Technology Cluster, Singapore University of Technology and Design (SUTD), Singapore, Singapore. [7] Department of Materials Science and Engineering, National University of Singapore, Singapore, Singapore. [8] Cancer Science Institute of Singapore, National University of Singapore, Singapore, Singapore. [9] Academic Clinical Program in Ophthalmology, Duke-NUS Medical School, Singapore, Singapore. [10] Department of Cellular, Developmental and Regenerative Biology, Icahn School of Medicine at Mount Sinai, New York, NY, USA. [11] Department of Physiology, Yong Loo Lin School of Medicine, National University of Singapore, Singapore, Singapore. [12] Department of Ophthalmology, National University Hospital, Singapore, Singapore. [13] These authors contributed equally: Bhav Harshad Parikh, Zengping Liu. ✉email: lohxj@imre.a-star.edu.sg; xysu@imcb.a-star.edu.sg

Aberrant wound healing, which leads to irreversible fibrosis, underpins many pathological diseases. In the eye, this manifests as proliferative vitreoretinopathy (PVR), a major cause of poor vision due to failed retinal detachment (RD) surgery[1,2]. PVR occurs due to proliferative and contractile fibrocellular scar membranes that form either in the vitreous or surrounding the retina. Formation of these fibrocellular membranous tissues, termed epiretinal membranes (ERMs) or sub-retinal membranes (SRMs), are driven primarily by retinal pigment epithelial (RPE) cells that have undergone epithelial-mesenchymal transition (EMT)[3]. These transformed RPE cells may acquire a migratory phenotype and form myofibroblast-like cells with contractile properties[4,5], resulting in recurrent tractional RDs. Clinically, PVR is a challenging complication to manage that requires complex surgical removal, with guarded prognosis for visual recovery[6,7].

PVR has a complex pathogenesis involving multiple pathways, including inflammation, fibrosis, and EMT[4]. Numerous cytokines such as tumour necrosis factor alpha (TNF-α), transforming growth factor beta (TGF-β), interleukin-6 (IL-6) are known to regulate pro-inflammatory and pro-fibrogenic responses in the eye[8]. In addition, transcription factors from the Snail and Twist family, but not the Nuclear factor erythroid 2-related factor 2 (NRF2) pathway, have traditionally been described to play a role in EMT within the eye[3,9]. Although many treatments targeting these pathways have been explored, there is no effective pharmacologic agent to date for the prevention or reversal of PVR[10,11]. Therein lies an unmet clinical need, to develop an approach for the prevention and treatment of PVR, and more broadly other types of aberrant wound healing.

Synthetic polymeric hydrogels are a class of biomaterials that are often used in various biomedical applications, because of their tunable and versatile physicochemical properties[12–14]. They can exist in various states when composed of amphiphilic blocks[11]. At low concentrations in aqueous solutions, amphiphilic polymers have the ability to spontaneously self-assemble into polymeric micelles once above their critical micelle concentration (CMC)[15–19]. Polymeric micelles have been widely used as drug delivery vehicles for various therapeutics due to their unique properties[20]. These include biocompatibility, bioavailability, drug loading capacities, large solubilisation capacity, and prolonged circulation time in vivo[21–24]. Beyond their function as mere physical drug carriers, artificially engineered polymeric materials have recently been shown to influence material-cell interactions just by tuning their physiochemical properties[25]. This includes the potential to influence cellular behaviour such as apoptosis, proliferation, and migration[26–28]. However, little is known about the underlying molecular mechanisms governing this cellular phenomenon, nor has it been harnessed for therapeutic applications.

In this study, we investigated if a particular multiblock thermo-responsive polymer, comprising poly(ethylene glycol) (PEG), poly(propylene glycol) (PPG), and poly(ε-caprolactone) (PCL), termed poly(CEP), is able to prevent aberrant wound healing in the eye. We demonstrate that the polymeric material itself is able to inhibit retinal scarring both in an in vivo experimental PVR rabbit model, as well as an in vitro induced RPE contraction model. This is mediated via the intracellular uptake of poly(CEP), leading to the activation of the NRF2 signalling pathway in RPE cells.

## Results

### Poly(CEP) prevents retinal scarring and contraction in rabbit experimental model of PVR. Poly(CEP) was synthesised and characterised as described (Supplementary Fig. 1 and Supplementary Table 1)[19,29]. A surgically induced experimental rabbit model of PVR was established to test the in vivo efficacy of poly(CEP) in preventing retinal scarring (Supplementary Movie 1). In this experimental PVR rabbit model, intravitreal co-injection of immortalised RPE cells (ARPE-19 cell line) and rabbit blood (Supplementary Fig. 2a) was performed, after core vitrectomy and induction of RD. Injection of RPE cells and blood has been reported to potentiate intraocular scarring through the formation of contractile fibrocellular membranes[30]. PVR severity in all experimental rabbits was graded (Supplementary Table 2). When air alone was injected to serve as a temporary vitreous tamponade ($n = 5$), retinal contraction and in-folding were observed at day 2 by infrared reflectance (IR) and spectral domain-optical coherence tomography (SD-OCT) imaging (Supplementary Fig. 2b). This progressed to RD by day 10 in all 5 cases (PVR grading of 5), whereby the detached retina was seen in the colour fundus and IR images (white arrows, Fig. 1a), and confirmed on SD-OCT. In eyes filled with 20% sulphur hexafluoride ($SF_6$, $n = 5$), a gas that is routinely used in the clinic as short-term vitreous tamponade[31], presence of ERMs with focal retinal traction (PVR grading of 2) was observed in 4 of 5 cases, with RD and contracted retina (PVR grading of 5) observed in the fifth case at 2 months. This is in contrast to 10 wt% poly(CEP)-filled eyes ($n = 8$), whereby the preservation of retinal structure on SD-OCT, and absence of ERM (PVR grading of 1) was observed in 7 of 8 cases, and PVR grading 2 was observed in the eighth case. In all 8 cases, the retinotomies healed and retina remained attached for up to 2 months post-surgery (Supplementary Fig. 2c–e). Taken together, these findings suggested poly(CEP)'s superiority in preventing PVR.

To ascertain if poly(CEP) is able to prevent both epiretinal and subretinal scarring in the rabbit retina, ex vivo haematoxylin and eosin (H&E) staining was performed at 2 weeks and 2 months post-surgery (Fig. 1b, c). Normal rabbit retinal structure with an intact underlying RPE was observed in all eyes filled with poly(CEP) only (without induction of PVR, $n = 5$ at 2 months, Supplementary Fig. 3), similar to non-operated control eyes ($n = 5$). In air-filled eyes ($n = 5$) which developed RD and contracted retina in all 5 cases, only a representative area with a partially attached retina is shown (Case #5). Here, the retina was self-folded and lacked distinct retinal layers. Multi-layered scar tissue was also observed subretinally, suggesting possible hyperproliferation and fibrosis. In experimental PVR rabbit eyes filled with poly(CEP) ($n = 8$), a normal retinal structure was observed for up to 2 months post-surgery, comparable to the non-operated control eyes. Most cases did not develop ERMs (6 of 8), and none developed SRMs. This is in contrast with $SF_6$-filled eyes ($n = 5$), whereby ERMs and SRMs were observed in 4 of 5 cases, leading to localised retinal traction and retinal distortion, both of which are known to precede PVR.

The presence of ERMs and SRMs was further characterised by immunofluorescence (IF) microscopy to visualise Alpha Smooth Muscle Actin (α-SMA) and Collagen Type I Alpha 1 Chain (COL1A1) (Fig. 1d, e), key fibrosis markers known to be highly expressed in membranes dissected from PVR patients[32]. In the non-operated eyes ($n = 5$), the retinal structure was well-maintained, with the absence of fibrotic markers. In the air-filled eyes ($n = 2$) at 2 weeks post-surgery, α-SMA and COL1A1 expression were present at both epiretinal and subretinal zones, indicative of severe fibrosis and presence of ERMs and SRMs. Furthermore, COL1A1 upregulation was observed intra-retinally, confirming the fibrotic response in the experimental PVR model[33]. In contrast, both markers were absent in the epi-, intra-, and subretinal zones of 10 wt% poly(CEP)-filled eyes at both 2 weeks (4 of 4 cases) and 2 months (4 of 4 cases). Consistent with the H&E findings, there was intense α-SMA and COL1A1 expression in both the epi-retinal and sub-retinal membranes of $SF_6$-filled eyes (3 of 3 cases) at 2 months. This

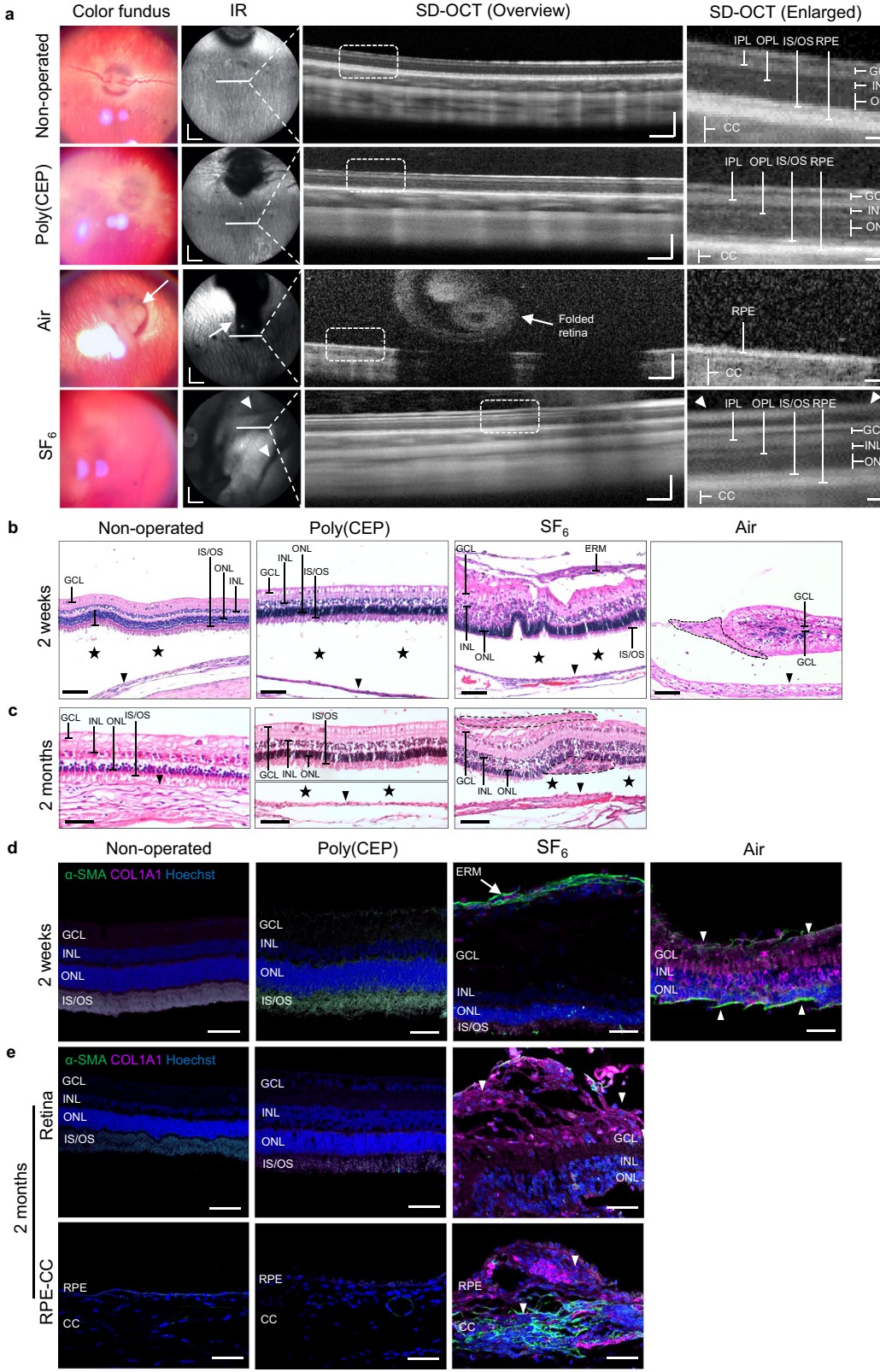

suggests that 10 wt% poly(CEP) was able to prevent the formation of aberrant epi- and subretinal fibrosis in a rabbit experimental model of PVR.

**Poly(CEP) hydrogel sheds polymeric micelles that are endocytosed by RPE cells**. Poly(CEP) undergoes reversible phase changes, from unimers to polymeric micelles as the concentration increases and from a solution to gel state as the temperature increases (Supplementary Fig. 4a). Poly(CEP) in the gel state, when injected into the vitreous cavity, primarily biodegrades via surface erosion and shedding of its component micelles over a period of 3 months, before it fully biodegrades via partial hydrolysis of ester bonds with PCL segments into its oligomers[14,29,34]. We hypothesised that micelles, shed from

**Fig. 1 Poly(CEP) prevents scarring and retinal detachment in an experimental rabbit model of PVR. a** Representative ophthalmic images of rabbit posterior eye segments from non-operated ($n = 5$), poly(CEP)- ($n = 8$) and SF$_6$-filled ($n = 5$) eyes at day 14 post-surgery, and air-filled eye ($n = 5$) (experimental PVR model) at day 10 post-surgery. *Colour fundus panel*: Both non-operated and poly(CEP)-filled eyes showed a normal clear fundus with attached retina. SF$_6$-injected eye showed a hazy fundus with blurred view of vessels. Air-filled eye showed a completely detached retina (white arrow). *IR panel*: Both non-operated and poly(CEP)-filled eye displayed normal posterior segment. SF$_6$-injected eye showed fibrotic scar membranes indicated by the streaks of dark shadows (white arrowheads). Air-filled eye showed a dark shadow (white arrow) that was caused by the detached retina. Scale bar, 2 mm. *SD-OCT (overview) panel*: All groups except air-filled showed attached retina. Scale bar, 200 μm. *SD-OCT (Enlarged from dotted box) panel*: Both non-operated and poly(CEP)-filled eyes displayed normal stratified retinal layers. SF$_6$-injected eye showed presence of scar membrane (white arrowhead). Air-filled eye showed a completely detached retina. Scale bar, 50 μm. **b** Histopathological characterisation of enucleated rabbit retinas from all four groups at time points mentioned in (**a**). Retina from non-operated ($n = 3$) and poly(CEP)-filled eye ($n = 4$) looked morphologically similar with the presence of distinct and organised retinal layers and a continuous monolayered RPE (black arrowheads). SF$_6$-injected eye ($n = 2$) showed ERM causing retinal traction. Air-filled eye ($n = 2$) showed a folded retina with disorganised retinal layers and a scar membrane (black dotted outline). The native RPE was multi-layered. **c** Histology of retinas at 2 months post-surgery showed that poly(CEP)-filled eye ($n = 4$) maintained retinal layer organisation seen in the non-operated eye. Retina of SF$_6$-filled eye ($n = 3$) had presence of ERM and SRM (black dotted outlines). For both **b**, **c** artefactual tissue separation was observed due to sample processing (black stars). Scale bar, 100 μm. **d** Retinal cross sections from all groups at 2 weeks post-surgery were immunostained with fibrosis markers, α-SMA, and COL1A1. Non-operated and poly(CEP)-filled eyes ($n = 4$) showed minimal staining of all markers throughout all retinal layers. SF$_6$-filled eyes ($n = 2$) showed intense staining of both α-SMA and COL1A1 in both the pre-retinal fibrotic membrane (white arrow). Air-filled eyes ($n = 2$) had significant intra-retinal upregulation of COL1A1 with co-localisation of α-SMA at the epiretinal and outer retinal layers (white arrowheads). **e** Retinal sections from poly(CEP)- ($n = 4$) and SF$_6$-filled ($n = 3$) eyes at 2 months post-surgery were immunostained with α-SMA and COL1A1. Both poly(CEP)-filled and non-operated eyes ($n = 3$) had minimal α-SMA and COL1A1 staining, unlike SF$_6$-filled eyes which had positive staining both at the epiretinal (top retina panel) and subretinal zones (bottom RPE-CC panel). Nuclei was stained in blue with Hoechst stain. Scale bar for **d**, **e**, 50 μm. CC = Choriocapillaris. ERM = epiretinal membrane. GCL = ganglion cell layer. INL = inner nuclear layer. IPL = inner plexiform layer. IS/OS = photoreceptor inner/outer segments. ONL = outer nuclear layer. OPL = outer plexiform layer. PVR = proliferative vitreoretinopathy. RPE = retinal pigment epithelium. SRM = subretinal membrane.

poly(CEP) hydrogel during in vivo biodegradation, may influence the biological function of both the injected ARPE-19 cells and native rabbit RPE, thereby preventing an aberrant fibrotic response and retinal scarring. Before testing this hypothesis, the CMC of poly(CEP) at body temperature (37 °C) in basal cell culture medium was determined to be 0.03 wt% (Supplementary Fig. 4b, c). Micellar size at various concentrations was quantified using dynamic light scattering. The average micelle size did not vary significantly between the lowest (0.05 wt%) and highest (1 wt%) concentrations measured (at 0.05 wt%: 153.20 ± 3.96 nm; at 1 wt%: 154.43 ± 14.11 nm) (Supplementary Fig. 4d). Poly(CEP) micelles have also been shown using small-angle X-ray scattering (SAXS) to undergo dynamic shape and morphological changes from spherical to cylindrical-shaped upon temperature changes[35].

To understand the underlying mechanism through which poly(CEP) micelles were able to prevent retinal scarring, we studied its interaction with human embryonic stem cell-derived RPE (ES-RPE) cells that closely resemble native human RPE cells[36]. To enable live in vitro intracellular tracking, ES-RPE cells were cultured and exposed to poly(CEP) conjugated with a fluorescein derivative (F-poly(CEP))[37]. ES-RPE cells were exposed to poly(CEP) micelles at concentrations below CMC (0.001 and 0.01 wt%) and above CMC (0.1, 0.5, and 1 wt%) for 4 h, and fluorescence from internalised poly(CEP) was detected by flow cytometry. There was a concentration-dependent increase in the mean fluorescence intensity (MFI) with a peak at 1 wt% (Fig. 2a), indicating that the intracellular uptake of poly(CEP) increases at higher concentration of poly(CEP) micelles.

Next, we studied the route by which F-poly(CEP) was internalised into RPE cells. Minimal uptake of 1 wt% F-poly(CEP) into ES-RPE cells (10.03 ± 6.70%, $P < 0.0001$) was observed at 4 °C when compared to uptake at 37 °C (92.24 ± 1.63%) (Fig. 2b). This confirmed that F-poly(CEP) was internalised into RPE cells by an active, energy-dependent uptake mechanism, rather than simple adhesion of micelles to the cell surface. To ascertain that F-poly(CEP) was taken up by endocytosis and to identify the endocytic pathway involved, inhibitors of clathrin-mediated endocytosis (chlorpromazine), caveolae-mediated endocytosis (genistein), micropinocytosis (amiloride) and lipid raft-

dependent internalisation (methyl-β cyclodextrin)[38,39] were used. Chlorpromazine inhibited cellular uptake the most with the lowest residual internalisation (26.22 ± 5.20%, $P < 0.0001$), followed by Genistein (69.26 ± 4.55%, $P < 0.0001$), suggesting that clathrin-mediated endocytosis was the predominant pathway for poly(CEP) internalisation. Post-internalisation trafficking was next analysed by exposing RPE cells to 1 wt% F-poly(CEP) in the presence of either LysoTracker (a marker for lysosomes), or Chloroquine (a lysosomotropic agent)[40] that inhibits lysosome function by increasing intra-lysosomal pH, and causes lysosome accumulation (Fig. 2c). 1 wt% F-poly(CEP) co-localised with LysoTracker at 24 h indicating its accumulation in lysosomes. This was prevented by the addition of chloroquine, consistent with endo-lysosomal trafficking of internalised poly(CEP) micelles in RPE cells.

**Internalised poly(CEP) micelles prevent contractile membrane formation in vitro.** To understand how the internalised poly(CEP) micelles were able to prevent scarring in vivo, we adopted a TNF-α and TGF-β1 (TNT)-induced RPE contraction model of PVR[41,42]. TNT treated RPE cells underwent EMT in four distinct stages to form contractile fibrocellular membranes over a 72 h period (Fig. 2d and Supplementary Movie 2). First, RPE cells acquired a mesenchymal-like appearance (Stage 1), followed by coalescing of the cells (Stage 2) to form ERM-like membranes (Stage 3), which then progressed to a contractile membrane (Stage 4) before eventually lifting off the tissue culture plate.

To recapitulate the in vivo scenario whereby 10 wt% poly(CEP) hydrogel was injected into the vitreous cavity, 10 wt% poly(CEP) was placed into the upper chamber of a dual chambered tissue culture system, separated by a porous polyester filter (pore size of 400 nm) (Supplementary Fig. 5a). The 400 nm pore size ensured that only individual polymeric micelles (size range: 139–154 nm), but not large micelle aggregates, diffuse from the upper to the lower chamber. This was further supported by a dye solubilisation assay experiment, whereby significant micelle shedding through surface erosion was confirmed at 48 h with a continuous increase until 96 h (Supplementary Fig. 5b–d).

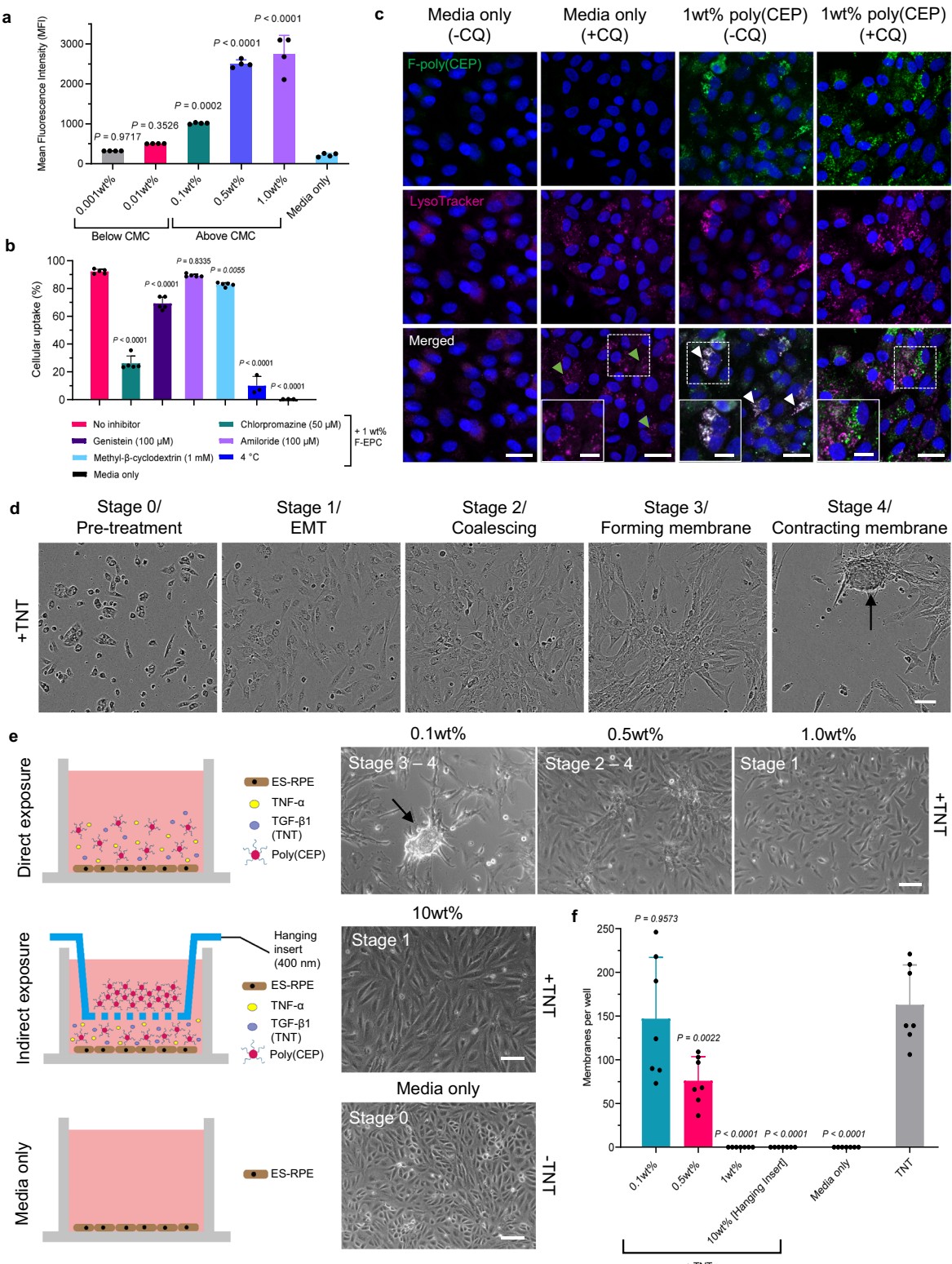

In this indirect exposure assay, micellar poly(CEP) was able to prevent the formation of contractile membranes and ES-RPE cells remained in Stage 1 (Fig. 2e). To further confirm that this phenomenon was due to intracellular uptake of poly(CEP), ES-RPE cells were directly exposed to three concentrations of poly(CEP) (0.1, 0.5, and 1 wt%), concentrations at which micelles are known to form and be internalised by cells (Fig. 2a). We observed a concentration-dependent suppression of

membrane formation. Partial inhibition of membrane formation occurred at 0.5 wt% (76.14 ± 27.41 membranes, $P = 0.0022$) and was completely inhibited at 1 wt% poly(CEP) + TNT (Fig. 2f and Supplementary Movie 3). This was consistent with previous findings showing that maximum internalisation of micelles was observed at 1 wt%, and confirmed that the anti-scarring effects were conferred by the micellar form of poly(CEP).

**Fig. 2 Intracellular uptake of poly(CEP) micelles in RPE cells prevent scar membrane formation. a** Mean fluorescence intensity (MFI) plot showed a significant increase in cellular uptake of fluorescein-conjugated poly(CEP) (F-poly(CEP)) at concentrations above the critical micelle concentration (CMC) (0.1, 0.5, and 1 wt%) when compared to media only at 4 h post-exposure. Data represents mean ± s.d. of four replicates. **b** Effect of different endocytosis inhibitors on cellular uptake (%) of 1 wt% F-poly(CEP). The greatest decrease in internalised F-poly(CEP) was observed after exposure to chlorpromazine, a clathrin-mediated endocytosis inhibitor. Minimal internalisation was observed when cells were kept on ice (4 °C). Data represents mean ± s.d. of three replicates for 4 °C and media only, and seven replicates for the remaining groups. **c** Intracellular uptake of 1 wt% F-poly(CEP) and its co-localisation with lysosome (marked using LysoTracker) was determined by immunofluorescence (IF) in the presence or absence of chloroquine (CQ), a lysosomotropic agent that prevents endosome to lysosome fusion. Addition of CQ increased punctate staining of LysoTracker (green arrows). 1 wt% F-poly(CEP) with LysoTracker showed the presence of internalised micelles co-localising with lysosomes (white arrows). However, the addition of CQ disrupted this co-localisation. Nuclei was stained in blue with Hoechst stain. Inset shows enlarged image from dotted box. Scale bar, main panel: 20 μm, inset: 10 μm. A representative image is shown from three independent experiments conducted. **d** Establishment of TNF-α and TGF-β1 (TNT) in vitro RPE cell contraction model of PVR. Time-lapsed phase contrast image frames over period of 72 h highlighting four progressive stages of RPE transformation after treatment with TNT. Black arrow points to scar membrane. Scale bar, 50 μm. **e** Poly(CEP) was exposed directly or indirectly to RPE cells as shown in the schematic. Indirect exposure of 10 wt% poly(CEP) prevented the progression of RPE cells beyond stage 1. Direct exposure of 1 wt% poly(CEP) prevented the progression of RPE cells beyond stage 1 in a concentration-dependent manner. Black arrow points to membrane. Scale bar, 100 μm. A representative image is shown from at least three independent experiments conducted. **f** The number of membranes per well for all groups in (**e**) were manually quantified. There was a significant but partial inhibition on membrane formation in 0.5 wt% poly(CEP) + TNT and complete prevention in both 1 wt% poly(CEP) + TNT and 10 wt% poly(CEP) + TNT. Data represents mean ± s.d. of seven replicates. Gating strategy for **a**, **b** is shown in Supplementary Fig. 11a, b, respectively. Statistical analyses for **a**, **b**, and **f** were performed using one-way ANOVA, followed by Tukey's honest significance difference (HSD) *post-hoc* test.

---

**Effect of poly(CEP) micelles on RPE cell proliferation and migration.** Hyper-proliferation contributes to retinal scar tissue formation[43]. Therefore, we studied the effect of poly(CEP) micelles on cell proliferation, by monitoring the expression of the proliferation marker Ki67[44] in RPE cells at three different concentrations of poly(CEP) (0.1, 0.5, and 1 wt%) using both the TNT-induced in vitro model of PVR (Fig. 3a) and exposure to poly(CEP) alone (Fig. 3b). The number of Ki67 positive nuclei in TNT-treated ES-RPE cells were reduced in the presence of 1 wt% poly(CEP), and this correlated with the absence of contractile membrane formation. Similarly, in the presence of poly(CEP) alone, a significant decrease in the mean fraction of Ki67 positive nuclei was seen in RPE cells exposed to 0.5 wt% (21.14 ± 4.41%, $P = 0.002$) and 1 wt% (19.39 ± 2.75%, $P = 0.0005$) poly(CEP) compared to media only control (35.78 ± 3.57) (Fig. 3c). Cell proliferation was also monitored using a growth curve, by measuring the change in mean confluence (% area occupied) over time (Fig. 3d). Consistent with the Ki67 data, both 0.5 and 1 wt% poly(CEP) alone were able to suppress RPE cell proliferation. Furthermore, lactate dehydrogenase (LDH) assays were performed to confirm that poly(CEP) at all concentrations (below and above CMC) was not cytotoxic to ES-RPE cells (Fig. 3e). Taken together, poly(CEP) was able to suppress proliferation both in the presence and absence of TNT in a concentration-dependent manner (i.e., above the CMC whereby micelles are formed) indicating that this is a micelle-specific phenomenon.

Migration of RPE cells to the epiretinal surface is a key step in ERM formation and PVR[45]. To determine if poly(CEP) suppresses ES-RPE cell migration, two concentrations of poly(CEP) (above its CMC; 0.1 and 1 wt%) were used, either in the presence or absence of TNT. Phase contrast images were taken at 12 h intervals up to 36 h (Fig. 3f) and wound closure was quantitated based on the remaining cell-free area at the different time points as a proxy for rates of cell migration (Fig. 3g). Attenuation of cell migration by poly(CEP) was observed only for the 1 wt% poly(CEP) alone group at 36 h (55.56 ± 7.86%, $P < 0.0001$), but not for 0.1 wt% poly(CEP) (73.72 ± 6.19%, *ns*), compared to media only control (75.06 ± 3.21%). Similarly, RPE cell migration was reduced only in the 1 wt% poly(CEP) + TNT group (78.22 ± 5.45%, $P = 0.0001$) but not in the 0.1 wt% poly(CEP) (92.33 ± 3.30%, *ns*) group, compared to TNT controls (96.37 ± 1.83%). This indicates that as expected ES-RPE migration was suppressed only at 1 wt% poly(CEP).

**Poly(CEP) suppresses EMT-related transcription factors.** In the TNT-induced RPE contraction model of PVR, 1 wt% poly(CEP) exposed cells appeared to adopt a mesenchymal-like morphology (stage 1), but did not progress to stages 2, 3, and 4 (Fig. 2e), suggesting incomplete EMT. Therefore, we investigated if intracellular uptake of poly(CEP) micelles suppressed gene expression of EMT-mediating transcription factors. Reverse transcription-quantitative polymerase chain reaction (RT-qPCR) and IF microscopy were performed for key EMT transcription factors and their downstream targets. ES-RPE cells treated with 1 wt% poly(CEP) + TNT lost their epithelial identity as shown by discontinuous cell borders demarcated by TJP1 (ZO-1) staining, a reduction in CDH1 (E-Cadherin) staining, and the loss of nuclear OTX2 labelling (Fig. 4a). Consistent with this, mRNA expression levels for key epithelial genes such as *Rpe65, Otx2, Tjp1,* and *Cdh1* were reduced by both TNT and 1 wt% poly(CEP) + TNT, indicating a partial loss of the epithelial phenotype (Supplementary Fig. 6).

ES-RPE cells treated with 1 wt% poly(CEP) + TNT had reduced expression of the EMT master regulator SNAI1 (Snail), and downstream mesenchymal proteins, such as COL1A1 and FN1 (Fibronectin) by IF microscopy, when compared to TNT alone (Fig. 4b). This was further corroborated by RT-qPCR, whereby the upregulation of EMT transcription factors, *Foxs1* ($P = 0.0016$,), *Snai1* ($P < 0.0001$) and *Snai2* ($P < 0.0001$) and downstream ECM genes, such as *Col1a1* ($P < 0.0001$) and *Fn1* ($P = 0.0002$), and junctional gene *Cdh2* ($P = 0.0002$) was observed at the mRNA level in the TNT group was suppressed by 1 wt% poly(CEP) (Fig. 4c). This suggests that the internalised polymeric micelles were able to impair the EMT signalling network.

**Poly(CEP) micelles activate the NRF2 pathway, which may mediate the anti-scarring effect.** To further elucidate the upstream molecular mechanisms underpinning the anti-scarring triggered by the intracellular uptake of poly(CEP) micelles, we performed RNA sequencing (RNA-Seq) on ES-RPE cells exposed to 1 wt% poly(CEP) in the absence or presence of TNT at 8 and 24 h. Triplicates from each condition showed high concordance ($r \geq 99\%$) in heatmap correlation and principal component analysis (PCA) plots (Supplementary Fig. 7a, b). PCA displayed distinct grouping of samples for each condition, indicating that there were unique transcriptomic profile changes in ES-RPE cells exposed to TNT alone, poly(CEP) alone or in combination by 8 h, that further amplified by 24 h. We used DESeq2 to identify the

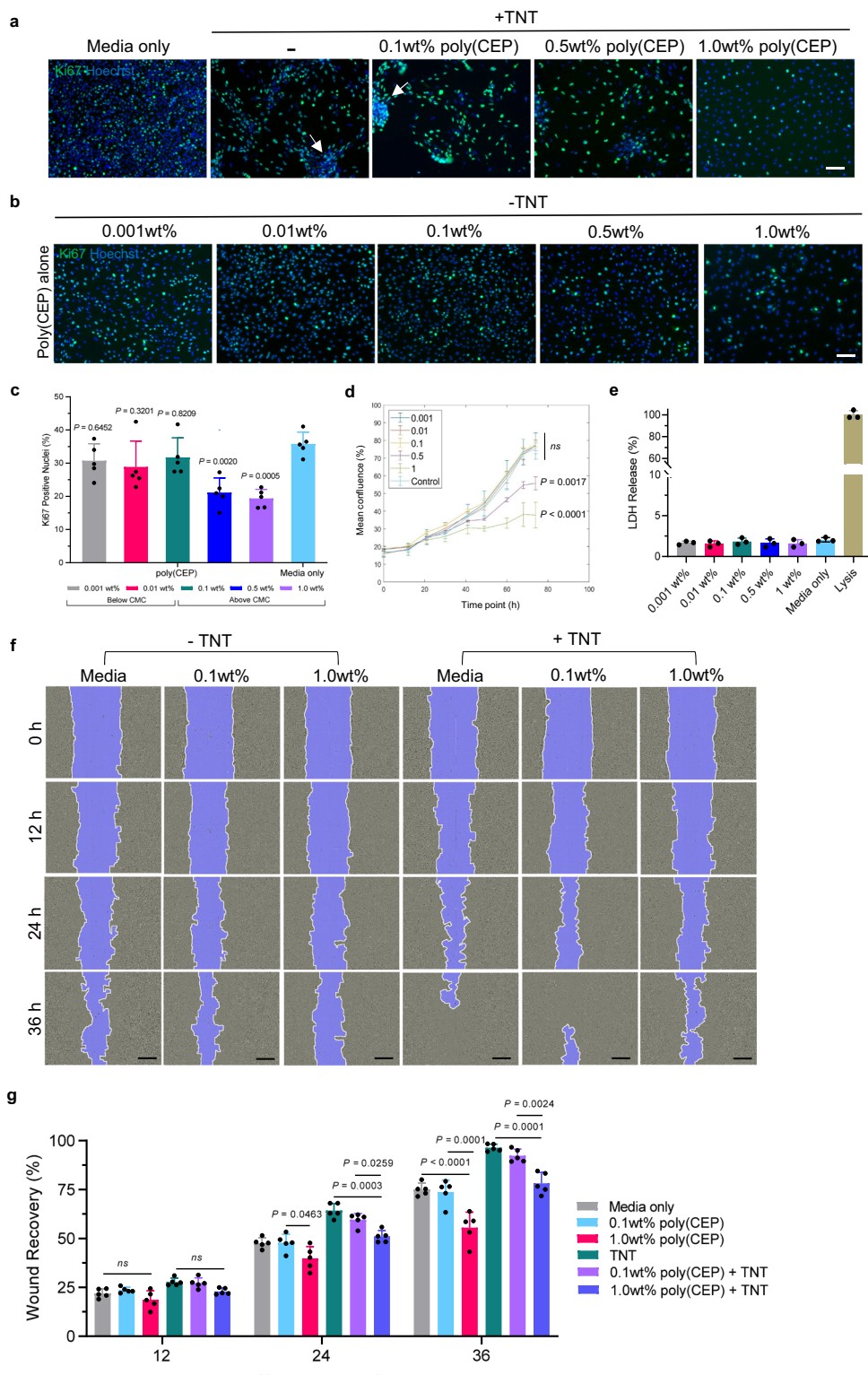

differentially expressed genes (DEGs) between the four conditions at both time points. Consistent with the PCA findings, these comparisons revealed significant changes in the expression level of 3781 and 5162 genes in poly(CEP) + TNT compared to TNT, and 3472 genes and 6644 genes in poly(CEP) alone compared to media only control at 8 and 24 h, respectively (Supplementary Fig. 7c, d, and Supplementary Data 1 for full DE genes of both comparisons, + time points). 62% of DEGs (i.e 1484 out of 2380) in the poly(CEP) + TNT group were also upregulated in the

poly(CEP) group (Fig. 5a and Supplementary Fig. 8a). This further corroborates our hypothesis that the phenomenon of in vitro PVR prevention is mediated primarily by poly(CEP) induced gene expression.

ES-RPE cells exposed to TNT were able to recapitulate PVR in vitro, as genes involved in the categories 'Epithelial to mesenchymal', 'Focal adhesion', 'Targets in ECM', 'Cell cycle' and 'Fibrosis' were upregulated and captured by gene set enrichment analysis (GSEA) at both 8 and 24 h (Fig. 5b,

**Fig. 3 Poly(CEP) suppresses RPE cell proliferation and migration in a concentration-dependent manner. a** Proliferation in RPE cells exposed to poly(CEP) micelles at 0.1, 0.5, and 1 wt% with TNT were ascertained by Ki67 IF. Only 1 wt% poly(CEP) visibly reduced the number of Ki67 positive nuclei. Membranes are indicated by white arrows. **b** Proliferation of RPE cells exposed to five concentrations (below CMC: 0.001 and 0.01 wt%; above CMC: 0.1, 0.5, and 1 wt%) of poly(CEP) without TNT were immunostained with Ki67. Nuclei was stained in blue with Hoechst stain. Scale bar for **a, b**, 100 μm. **c** All five concentrations in (**b**) and media only control from (**a**) were quantified for co-localisation of Ki67 with nuclei (using Hoechst stain). This co-localisation was plotted as the Ki67 positive nuclei (%) indicating actively proliferating cells at the endpoint of 72 h. Only poly(CEP) ≥ 0.5 wt% showed significant inhibition of proliferation compared to media only control. Data represents mean ± s.d. of five replicates. **d** Cell proliferation was monitored by plotting a curve using the mean area confluence (%) of cells imaged at regular intervals (10–12 h) exposed to five concentrations of poly(CEP) alone. Only poly(CEP) at ≥ 0.5 wt% showed significant suppression of proliferation compared to control (Media only). Data represents mean ± s.d. of three replicates. **e** Cytocompatibility of poly(CEP) at the same concentrations was studied using cytotoxicity lactate dehydrogenase (LDH) assay. LDH release (%) was negligible at all concentrations of poly(CEP) and media only control. Lysis buffer (provided by the manufacturer) was used as a positive control. Data represents mean ± s.d. of three replicates. **f** Scratch-wound healing assay was conducted to study RPE cell migration. Cells were exposed to two concentrations of poly(CEP) (0.1 and 1 wt%), with or without TNT. Phase contrast images of all groups were captured at four time points (0, 12, 24, and 36 h) with scratch-wound mask in blue. Scale bar, 300 μm. **g** Wound recovery (%) was quantified at three time-points (12, 24, and 36 h) post-scratch. Only poly(CEP) at 1 wt% displayed significant attenuation of migration compared to media only control at the endpoint of 36 h. This attenuation was also observed for 1 wt% poly(CEP) + TNT compared to TNT. Data represents mean ± s.d. of five replicates. Statistical analyses were performed for **c–e** and **g** using one-way ANOVA, followed by Tukey's honest significance difference (HSD) post hoc test. *ns*, not significant.

Supplementary Fig. 8b and Supplementary Data 2). Treatment of ES-RPE cells with poly(CEP), in both the presence and absence of TNT, downregulated genes involved in the categories 'EMT/ Migration', 'Cell cycle' and 'ECM' (Fig. 5c, d, Supplementary Fig. 8c, d, and Supplementary Data 3) consistent with our previous in vitro findings.

Addition of poly(CEP) micelles to both the TNT (PVR model) and non-TNT (media only) groups, triggered the upregulation of the NRF2 pathway compared to media alone. The NRF2 pathway is known to trigger the first line of homoeostatic response to a plethora of external and endogenous stimuli, through the activation of various downstream pathways[46]. To understand which pathways were being regulated by NRF2, we identified at least 40 NRF2 target genes that were significantly upregulated by 8 h and further amplified at 24 h (Fig. 6a, b and Supplementary Fig. 9), suggesting that the NRF2 pathway might be an early sensor of intracellular uptake of poly(CEP) micelles. These upregulated genes belong to the 8 different categories of NRF2-driven functions as indicated[46–49]. Upregulation of 16 NRF2 target genes from the various functional categories were validated via RT-qPCR and showed a positive correlation with RNA-Seq fold change ($r > 0.9$) (Fig. 6c, d and Supplementary Table 3). In a subset of these genes, expressions were quantified at earlier time points, indicating significant upregulation at 4 h (Fig. 6e). Consistent with this, increased nuclear localisation of NRF2 was observed in ES-RPE cells treated with poly(CEP) at 4 h (Fig. 6f, g). To further ascertain if poly(CEP)-dependent NRF2 activation occurred in vivo, we performed RNA-Seq on isolated retina and RPE-choroid tissues at 1-month ($n = 3$) and 2-month ($n = 3$) from a poly(CEP)-filled rabbit eye (without PVR induction). Congruent with observed in vitro activation of NRF2 target genes, both retina and RPE-choroid tissues also showed upregulation of genes from NRF2 pathway at both timepoints (Supplementary Fig. 10). Next, to determine if NRF2 activation was sufficient to elicit the anti-scarring effect in vitro, dimethyl fumarate (DMF), a pharmacological activator of NRF2[50], was added to ES-RPE cells in the presence of TNT (Fig. 6h). Activation of NRF2 by DMF was able to completely prevent contractile membrane formation, similar to poly(CEP) alone. Altogether, this suggests that NRF2 acts as an early sensor of poly(CEP) micelle intracellular uptake and master regulator of various downstream signalling pathways to confer poly(CEP)'s anti-scarring properties.

## Discussion
Synthetic polymeric materials have traditionally been viewed as mere inert physical carriers for the delivery of bioactive compounds to elicit specific biological effects[51]. However, emerging studies have suggested that polymeric materials themselves are able to directly influence cell behaviour[25,52–54]. In this study, we provide evidence using both an in vivo experimental rabbit model and an in vitro RPE contraction model of PVR, that poly(CEP) micelles have intrinsic anti-scarring properties, to prevent retinal scarring without adjunct pharmacologic agents. This is achieved via cellular internalisation of poly(CEP) micelle, activation of NRF2 mediated pathways, leading to impairment of EMT, and suppression of cell hyper-proliferation and migration.

Pharmacologic agents to prevent PVR have been explored widely. However, to date, none have been approved for clinical use. Here, we propose an approach for the prevention of a post-surgical scarring complication using only a synthetic material, poly(CEP). The polymer's efficacy in preventing both epi- and subretinal membranes in rabbit eyes were studied in comparison to $SF_6$, a common expansile gas used in the clinic for the management of RDs[31]. Although both poly(CEP) and $SF_6$ act as effective endotamponade agents to facilitate re-attachment of retina, only poly(CEP) was able to prevent both ERM and SRM formation in vivo. These membranes formed in $SF_6$ treated rabbit eyes were positive for key fibrosis markers α-SMA and COL1A1, known to be upregulated in PVR pathogenesis[32]. In contrast, these fibrosis markers were absent from the poly(CEP)-filled eyes.

Poly(CEP) suppresses proliferation, migration and EMT in RPE, the predominant cell type in the formation of contractile scar membranes[55,56]. We recapitulated the three pathological hallmarks of PVR in vitro using ES-RPE cells in a TNT-induced model of RPE contraction[41,42]. Our findings indicated that poly(CEP) alone is able to suppress the upregulation of canonical EMT regulators (*Foxs1*, *Snai1*, and *Snai2*) and their downstream targets (*Col1a1*, *Fn1*, and *Cdh2*), and attenuate migration and hyper-proliferation of RPE cells. This correlates well with the in vivo findings, whereby poly(CEP)-filled eyes in experimental PVR rabbit models, did not develop epiretinal or subretinal membranes.

To gain insight into molecular mechanisms of how 1 wt% poly(CEP) micelles may be modulating RPE cellular behaviour, genome-wide transcriptomic profiling was performed. Consistent with the in vitro data, intracellular uptake of poly(CEP) down-regulated expression of genes involved in cell proliferation, EMT, and migration pathways. Interestingly, a xenobiotic-related NRF2 pathway was identified to be consistently upregulated in the presence of poly(CEP). NRF2 is a pleiotropic transcription factor, which is known to be the master regulator of redox homoeostasis that activates cytoprotective genes and is an emerging target

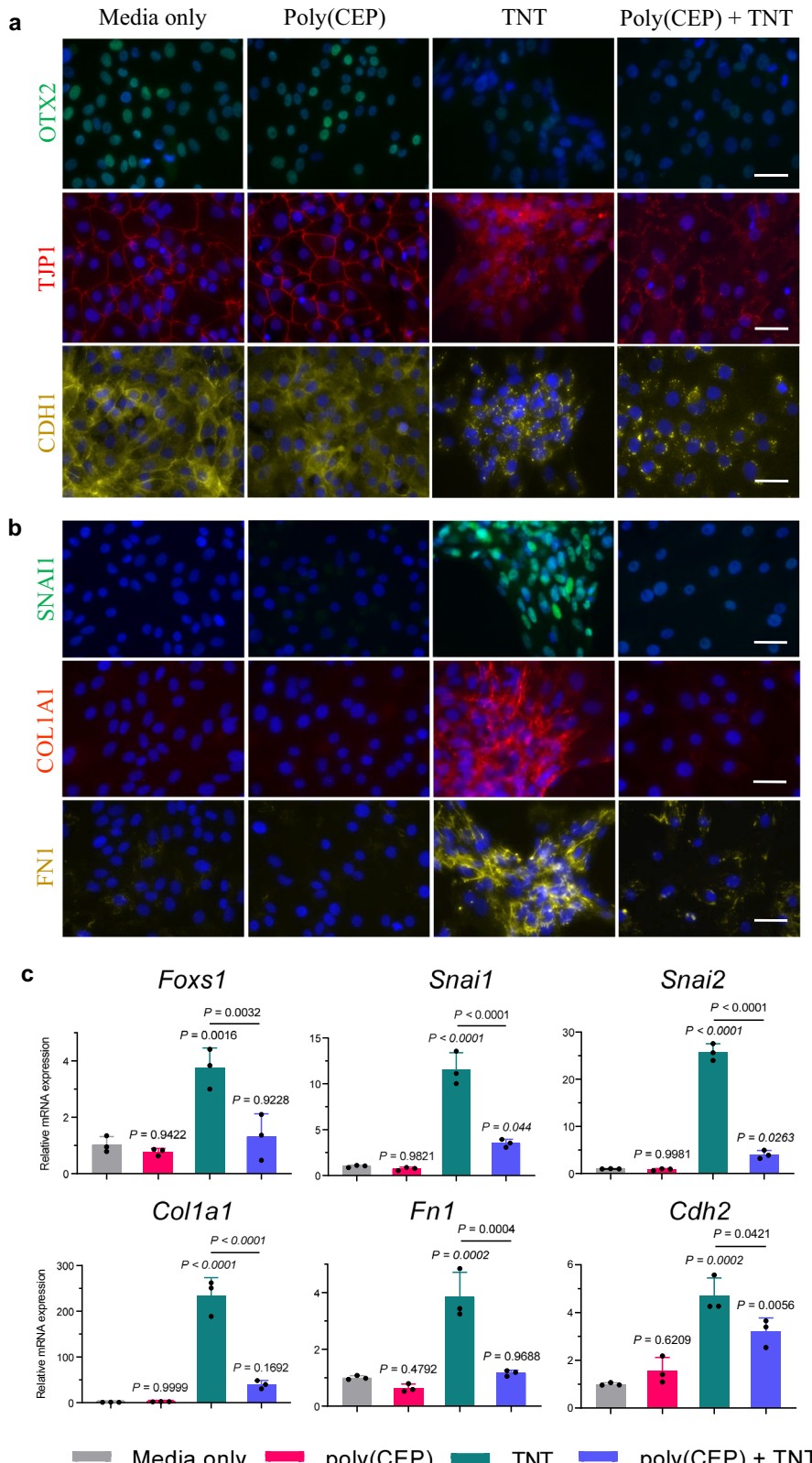

against various diseases such as neurodegeneration and cancer[57,58]. Although NRF2 has been suggested to regulate EMT, migration, and proliferation in some cell types[9,59], this has not been previously described in the context of PVR or RPE response to synthetic polymeric micelles. Increased nuclear localisation of NRF2 and early upregulation of its target genes at 4 h upon poly(CEP) exposure further confirmed NRF2 as one of the key

sensor and transcription regulator in response to external xenobiotic stimuli. Pharmacologic activation of NRF2 by DMF, (an FDA-approved drug)[50], was shown to be sufficient to prevent contractile membrane formation in vitro, albeit less effectively than poly(CEP) treatment alone. This suggests the presence of alternative pathways activated by poly(CEP) micelles that act synergistically with NRF2 to elicit cellular anti-scarring. Indeed,

**Fig. 4 Poly(CEP) at 1 wt% impairs EMT to inhibit membrane formation. a** RPE cells treated with TNT with or without poly(CEP) for 72 h were immunostained with RPE-specific transcription factor: OTX2, and cell junctional complex proteins: TJP1 (ZO-1) and CDH1 (E-Cadherin). Mislocalisation of CDH1, TJP1, and reduction of OTX2 observed in the presence of TNT, was partially reversed for TJP1 in the presence of 1 wt% poly(CEP). **b** Immunostaining with EMT transcription factor (TF): SNAI1 (Snail), and ECM proteins: COL1A1 and FN1 (Fibronectin) showed upregulation with TNT. 1 wt% poly(CEP) + TNT protected the cells against this upregulation. Nuclei was stained blue with Hoechst stain. Scale bar for **a**, **b**, 20 μm. A representative image is shown for **a**, **b** from three independent experiments conducted. **c** EMT TFs (*Foxs1*, *Snai1*, and *Snai2*) and key downstream mesenchymal genes (*Col1a1*, *Fn1*, and *Cdh2*) were quantified by quantitative PCR (qPCR). TNT induced upregulation of EMT TF was inhibited in the presence of 1 wt% poly(CEP). Data represents mean ± s.d. of three replicates for RT-qPCR. Statistical analysis was calculated using one-way ANOVA, followed by Tukey's honest significance difference (HSD) post hoc test. *ns*, not significant.

poly(CEP) has broad biological effect on RPE cells, as evidenced by the activation of diverse pathways identified in our in vitro RNA-Seq analysis. Activation of NRF2 pathway by poly(CEP) was further corroborated by our RNA-Seq analysis of rabbit RPE-choroid and retina tissues exposed to poly(CEP) in vivo. Further investigation is required to unravel the full repertoire of transcription factors and regulatory factors involved.

Interestingly, this anti-scarring effect by poly(CEP) was elicited in its micellar, but not unimeric form, and was mediated mainly through clathrin-dependent internalisation into ES-RPE cells. Both the internalisation of micelles, and its subsequent efficacy in inhibiting contractile scar membrane formation was concentration-dependent, with 1 wt% being the optimal concentration for poly(CEP)'s anti-scarring property. Physicochemical properties of micelles have been shown to change with concentration, such as reversibly changing from spherical to cylinder rods[14,35]. Our work suggests that changes in micelle structure affects cellular internalisation. Future work will focus on such rational engineering of physicochemical properties in these polymeric micelles, such as shape, surface charge, surface hydrophobicity/hydrophilicity, surface functionalization and stability, to elicit polymer specific cellular responses with therapeutic potential.

In conclusion, this study demonstrates that poly(CEP) is able to continuously shed micelles via surface erosion to suppress aberrant EMT in RPE cells after retinal surgery. This is the first report, whereby a synthetic polymeric material alone can target intracellular signalling pathways to prevent retinal scarring. More importantly, this study offers insight into how synthetic polymeric materials no longer function merely as inert drug carriers and challenges the conventional belief that a small molecule (drug) is always required to achieve a therapeutic effect at a cellular level. It highlights the potential of next generation nanomedicine, whereby polymers alone can be used to elicit specific biological responses. Lastly, this unique anti-scarring effect of poly(CEP) may be applicable for other clinical applications beyond ophthalmology, for example as an adjuvant treatment in cancer to prevent cell migration and metastasis.

## Methods

**Materials.** PEG with $M_n$ 2050 g/mol, PPG with $M_n$ 2000 g/mol, PCL-diol with $M_n$ 2000 g/mol, 1,6-hexamethylene diisocyanate (HMDI) (98%), dibutyltin dilaurate (DBTL) (95%), anhydrous toluene, and diethyl ether were purchased from Sigma-Aldrich. All reactants were used as received.

**Molecular characterisation.** Gel permeation chromatography (GPC) was performed with a Viscotek GPC max module equipped with two pheonogel columns ($10^3$ and $10^5$ Å) (size: 300 × 7.80 mm) in series and a Viscotek refractive index detector. Tetrahydrofuran (THF) was used as the eluent at a flow rate of 1.0 ml/min at a column temperature of 40 °C. Mono-dispersed polystyrene standards were used to obtain a calibration curve. Nuclear magnetic resonance (NMR) spectra were obtained at room temperature (RT), with the samples in CDCl₃ solvent, using a JEOL 500 MHz NMR spectrometer.

**Synthesis of poly(PEG/PPG/PCL) (poly(CEP)).** The thermogelling multiblock polymer poly(CEP) was synthesised from macromonomer-diols (PEG, PPG, and PCL-diol). PEG (12.00 g), PPG (3.00 g), and PCL-diol (0.15 g) were dried via azeotropic distillation before being polymerised by 1.22 mL ($7.58 \times 10^{-3}$ mol) of HMDI in the presence of DBTL ($\sim 8 \times 10^{-3}$ g) at 110 °C in argon atmosphere for 24 h. The polymerisation was initiated in 10 mL of anhydrous toluene and additional 10 mL of anhydrous toluene was introduced whenever the reaction mixture gelled with the final volume topped up to 100 mL by 4 h. The resultant poly(CEP) polymer was precipitated using diethyl ether to yield the crude polymer as a white solid. For in vitro and in vivo experiments, the poly(CEP) polymer was further purified via dialysis. The precipitated polymer was dissolved in isopropanol (IPA) (5 g of polymer in 50 mL IPA) and dialysed against deionised water using dialysis membrane (3.5 kDa MWCO). The deionised water was changed every 4 h for 3 days and the resultant purified polymer solution was frozen and lyophilised (purified yield = 72%). ¹H NMR characterisation (Supplementary Fig. 1b) shows that the molar ratio of PEG:PPG:PCL in synthesised poly(CEP) was 4.27: 1.00: 0.0647 (Supplementary Table 1). The synthesised poly(CEP) has $M_n \approx 26700$ g/mol, $M_w \approx 38800$ g/mol, and PDI ≈ 1.45 (GPC data in Supplementary Fig. 1c).

**Animals.** Twenty-one male New Zealand White rabbits aged 4–6 months and weighing 2.5–3.0 kg were purchased from BioSystems (BioSystems Corporation Pte Ltd., China). All animals were handled in accordance with the Statement of the Association for Research in Vision and Ophthalmology (ARVO) for the Use of Animals in Ophthalmic and Vision Research. All animal experiments were reviewed and approved by the Institutional Animal Care and Use Committee (IACUC) of SingHealth (Singapore) under protocol number 2015/SHS/1092 and 2019/SHS/1457. The animal facility at SingHealth Experimental Medicine (SEMC) is Association for Assessment and Accreditation of Lab Animal Care (AAALAC) approved. This ensures that all animal experimentation complies with standards as per National Advisory Committee for Laboratory Animal Research (NACLAR) guidelines set out by Agri-Food & Veterinary Authority (AVA) of Singapore.

**Animal study**
*Experimental PVR rabbit model.* A surgically induced experimental PVR model was established in rabbit eyes. A three-port, 25-gauge (G) core vitrectomy was performed in rabbits under general anaesthesia by intramuscular injection of ketamine (50 mg/kg, body weight (BW)) and xylazine (10 mg/kg, BW)[29,60]. Then, the posterior vitreous detachment was induced with the assistance of an intravitreal injection of triamcinolone acetonide. Localised RD (about 2–3 disc diameter) was induced by subretinal injection of approximate 30–40 μL balanced salt solution (BSS), followed by a 1 mm retinal break created by vitrectomy cutter. After air-fluid exchange, the vitreous cavity was co-injected with premixed 0.05 mL BSS containing 500,000 ARPE-19 (ATCC; CRL-2302) cells and 0.05 mL rabbit blood taken prior to the surgery. Finally, 10 wt% poly(CEP) ($n = 8$) or isoexpansile gas, 20% Sulphur hexafluoride (SF₆, 1:5 diluted with clean air, $n = 5$), was injected to replace the air in the vitreous cavity, while remaining cases filled with air ($n = 5$) served as temporary vitreous tamponade for PVR. Eyes ($n = 8$) filled with poly(CEP) gel after vitrectomy, without PVR model creation (inducing RD, co-injection of ARPE-19 cells and rabbit blood), served as an additional surgical operation control. Topical antibiotic ointments (Tobradex®, Tobramycin and Dexamethasone, Alcon) were applied to the treated eyes twice a day for 5 days post-surgery. In vivo live ophthalmic imaging follow-ups were carried out every other day post-surgery for 14 days for all cases, then once a week till 8 weeks (details summarised in Supplementary Table 2). Overview fundus images were taken with a 55° lens under infrared reflectance (IR) mode using Spectralis® SD-OCT device (Heidelberg Engineering). The SD-OCT images were taken with a 30° lens under IR + OCT mode at representative positions with the same device. Colour fundus images were taken with a 90D (Diopter) non-contact slit-lamp lens (Volk Optical Inc.) in front of a slit-lamp equipped with a digital camera (Righton).

*PVR severity grading.* The severity of PVR in all operated eyes was graded using a published grading scheme[61,62], on a scale of 0–5 based on representative fundus photos and SD-OCT images by two independent graders. In brief, stage 0: represented normal; stage 1: intravitreal membranes; stage 2: focal traction; stage 3: localised medullary ray detachment; stage 4: total medullary ray detachment and peripheral RD; stage 5: total RD and retinal folds (gradings are summarised in Supplementary Table 2).

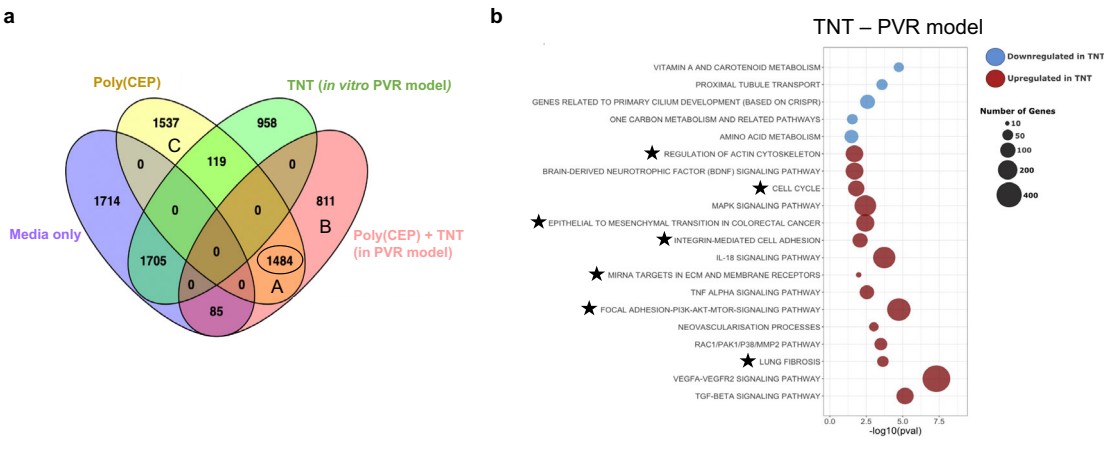

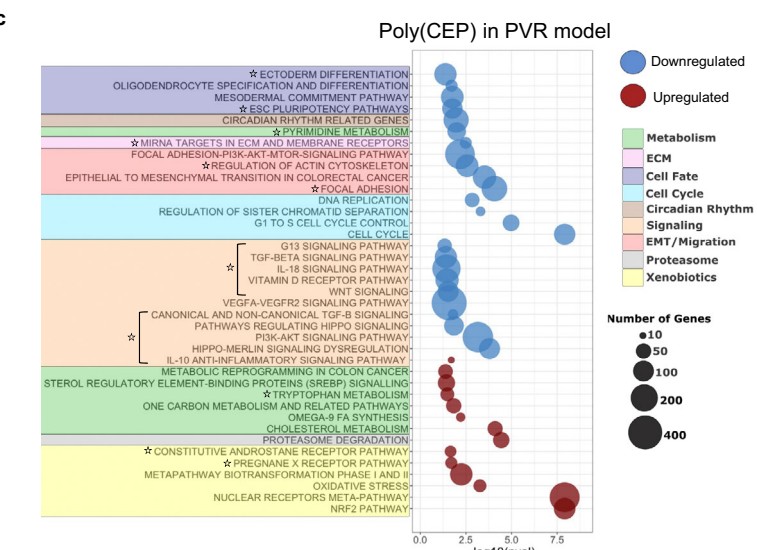

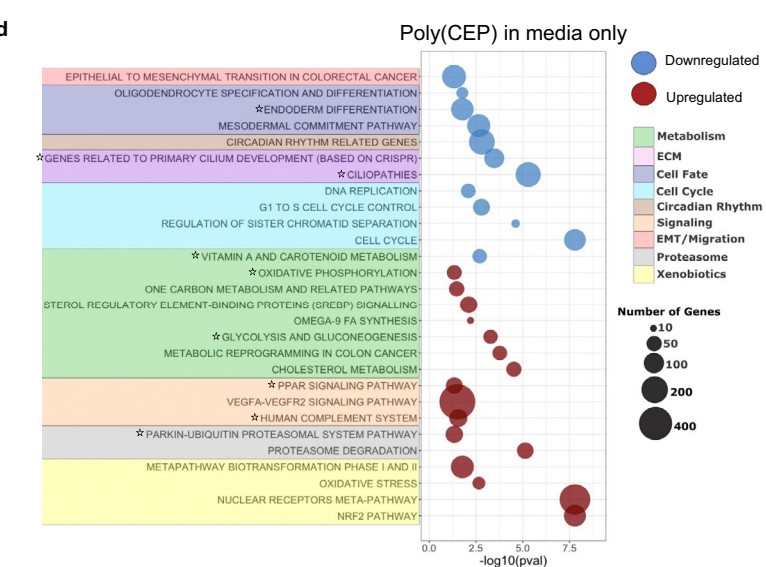

**Fig. 5 Intracellular uptake of 1 wt% poly(CEP) induces global transcriptomic changes in RPE. a** Venn diagram showing overlap of differentially expressed genes (Wald test, $P_{adj}$ < 0.05) identified by DESeq2 at 24 h post-exposure to media only, poly(CEP), TNT or poly(CEP) + TNT. 1484 genes are common in both poly(CEP) and poly(CEP) + TNT treated RPE cells. **b** Gene set enrichment analysis (GSEA) was performed on the in vitro PVR model induced by TNT at 24 h post-exposure and showed upregulation of classical pathways (Hollow black stars) with known involvement in PVR pathogenesis. **c** GSEA was performed for 1 wt% poly(CEP) + TNT (PVR model) 24 h post-exposure and showed downregulation of EMT, migration, and cellular proliferation pathways, but upregulation of pathways involved in xenobiotics, proteosome degradation, and metabolism, similar to **d** 1 wt% poly(CEP) in media only. Hollow black stars mark unique pathways between (**c**, **d**).

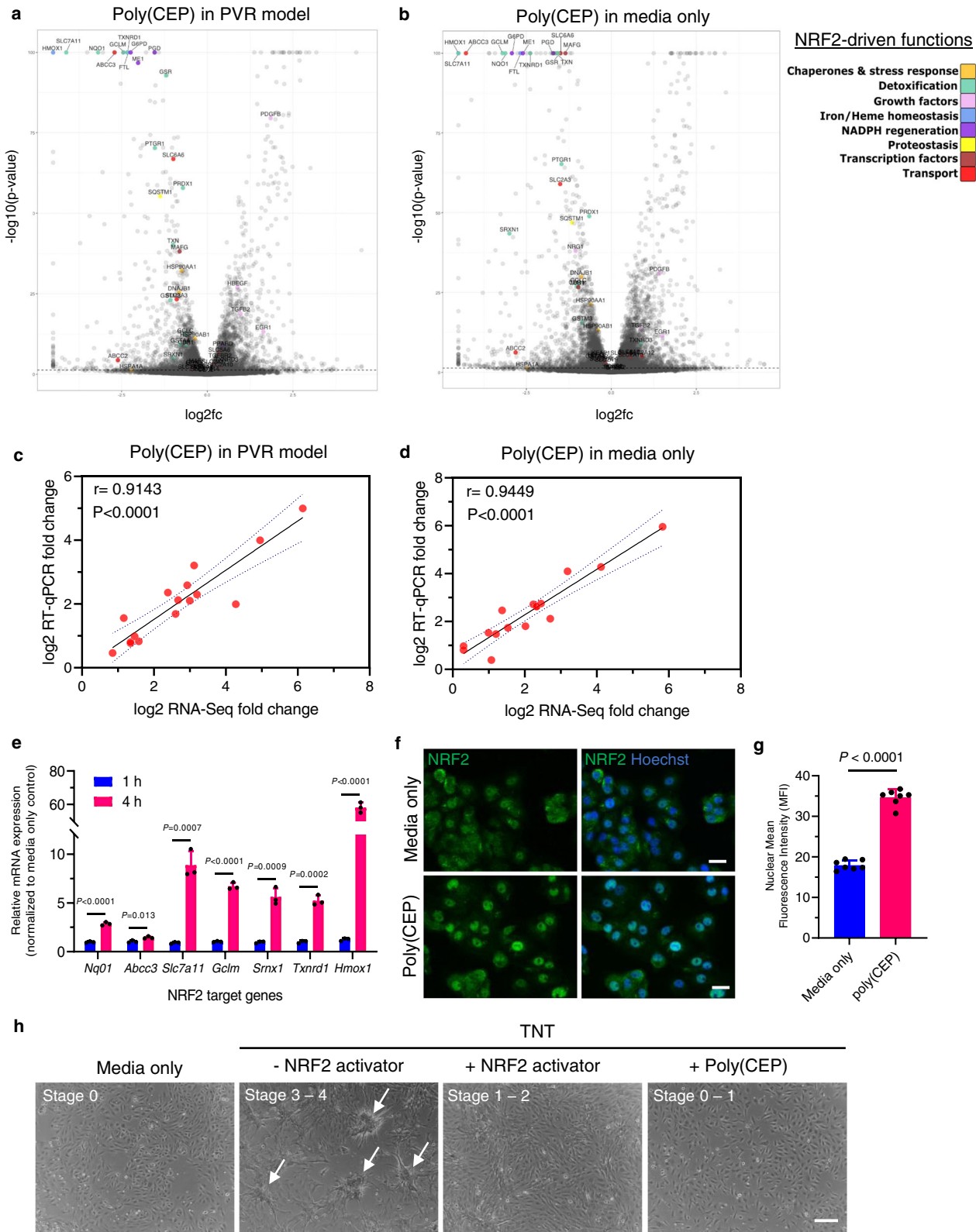

**Histopathological Examination**. Animals were euthanised in deep anaesthesia with an intracardiac injection of the euthanising agent (Pentobarbital). Following perfusion via the carotid artery with 10% neutral buffered formalin (Leica Biosystem)[63], the eyes were enucleated, and entire globes were immersed in 10% formalin for 48 h at 4 °C. The whole eyes were then embedded in paraffin. Sections were cut at 10 μm thickness with a microtome (Leica Biosystem) and further stained with haematoxylin (Merck Millipore) and eosin (Merck Millipore) (H&E). Light micrographs were taken on Zeiss Axio Imager. A2 microscope equipped with a digital camera.

**Immunofluorescence (IF) of paraffin-embedded eye sections**. Paraffin-embedded sections (10 μm) were deparaffinised in xylene followed by rehydration through a series of graded ethanol concentrations. The sections were subjected to antigen retrieval before proceeding with IF. They were blocked in 10% animal serum in phosphate-buffered saline (PBS) with 0.1% Tween-20 and 1% bovine serum albumin (BSA; Sigma-Aldrich) for 1 h at RT and incubated with primary antibodies overnight at 4 °C. The following primary antibodies were used, mouse anti-α-SMA and rabbit anti-COL1A1 (Supplementary Table 4). Secondary antibodies diluted in 0.1% PBS with 0.1% Tween-20 were added for 30 min at RT. The

**Fig. 6 Poly(CEP) prevents scarring via the activation of NRF2 signalling pathway. a, b** Volcano plot identified >40 genes in NRF2 signalling cascade (highlighted in colour) that were significantly upregulated at 24 h after poly(CEP) exposure, in the presence or absence of TNT. *P* values were calculated using the Wald test. The Benjamini-Hochberg (BH) method was used to adjust *p*-values for multiple comparisons. Dashed line indicates $P_{adj}$ = 0.05. **c, d** A positive correlation (*r* = 0.9143, *P* < 0.0001) was observed for 16 selected genes (subset of genes in (**a, b**)) between log2 RT-qPCR fold change and log2 RNA-Seq fold change values (details in Supplementary Table 3). **e** NRF2 target genes were quantified at the mRNA level (normalised to time-matched media only control) at 1 and 4 h post-exposure to 1 wt% poly(CEP) alone. There was a significant increase in expression of all selected NRF2 target genes at 4 h. Data represents mean ± s.d. of three replicates. **f** Nuclear localisation of NRF2 protein in RPE cells was ascertained by IF after 4 h exposure to 1 wt% poly(CEP) compared to media control. Nuclei stained in blue with Hoechst stain. Scale bar, 10 μm. **g** The nuclear localisation of NRF2 was quantified and presented as MFI. Cells exposed to 1 wt% poly(CEP) showed increased nuclear localisation of NRF2. Data represents mean ± s.d. of 7 replicates. **h** Dimethyl fumarate (DMF, 25 μM), a NRF2 activator, or poly(CEP) was added to the TNT-treated RPE cells. Both showed inhibition of contractile membrane formation. Status of cell stage of the RPE cell transformation is given in the panel. Scale bar, 200 μm. A representative image is shown from at least three independent experiments conducted. Statistical analyses were performed for **c, d** using two-tailed Pearson's correlation, **e, g** using two-tailed Student's unpaired *t* test.

secondary antibodies used were Alexa Fluor 488 donkey anti-mouse IgG and Alexa Fluor 568 donkey anti-rabbit IgG (Supplementary Table 4). A secondary antibody only negative control was included where sections were only incubated with blocking buffer in place of primary antibodies. Nuclei were stained with Hoechst 33342 (Thermo Fisher Scientific). Sections were mounted using ProLong Gold Antifade (Thermo Fisher Scientific) and stored at 4 °C until imaging. Images were taken using LSM800 confocal laser microscope (Zeiss).

**Poly(CEP) micelle characterisation**

*Critical micelle concentration*. The CMC of poly(CEP) polymer was determined by a dye solubilisation method. 1,6-diphenyl-1,3,5-hexatriene (DPH) was dissolved in methanol to form a solution at 0.6 mM. Each polymer was prepared at 10 concentrations ranging from 0.001 wt% to 1.0 wt% with 4 mL of basal media per concentration. 40 μL of 0.6 mM DPH solution was added to each 4 mL polymer solution and samples were equilibrated at RT in dark for 24 h. The absorbance spectra of the samples were recorded using a Shimadzu UV-2501 PC UV–VIS Spectrophotometer (Kyoto, Japan) in the range of 320–420 nm at 37 °C. The difference in absorbance at 376 and 400 nm was plotted against concentration (wt%). The CMC value of poly(CEP) was determined by the intersection of the extrapolation of fitted linear curve of unimeric and micellar regions.

*Micelle size*. Average size (Z-avg) of poly(CEP) micelles with respect to concentration (wt%) dissolved in basal media was measured using dynamic light scattering (Malvern Panalytical, Zetasizer Nano ZS). Micelle solution was prepared at 2 wt% and equilibrated at 37 °C overnight before being subjected to serial dilution. Size of poly(CEP) micelles at each concentration was measured 5 min after dilution while being incubated at 37 °C.

*Micelle shedding from hanging insert*. 150 μL of poly(CEP) micelle solution at each concentration was aliquoted into a 96-well plate and 15 μL of 0.6 mM DPH solution was added to each well. The solutions were allowed to equilibrate overnight at RT in dark. A standard curve of poly(CEP) concentration (wt%) against difference in DPH absorbance (377–396 nm) was fitted with a quadratic trend line; the equation of the trend line is as shown in the graph and the $R^2$ is 0.9743. 0.1 g of 10 wt% poly(CEP) (dissolved in AMO® ENDOSOL BSS, Abbott Medical Optics) was loaded into hanging inserts (400 nm pore diameter) and the exterior well filled with 900 μl basal media. Shed micelles were collected from the exterior well and plated in triplicates followed by addition of 0.6 mM DPH solution. Shed micelle concentration was quantified by comparing their difference in DPH absorbance (377–396 nm) with the standard curve obtained. Graph was plotted showing concentration of shed micelles (wt%) against duration (h).

**Embryonic stem cell-derived RPE cell culture (ES-RPE)**. ES-RPE cells were derived from human embryonic stem cell line H9 ES cells (WA09, Wicell) and differentiated according to published protocol with minor modifications[64]. In brief, ES cells were seeded in six-well culture plates pre-coated with Matrigel® (Corning) and grown in mTesR® medium (Stem Cell Technologies) to 90% confluency before commencing differentiation. At the end of differentiation (Day 18), the medium was switched to RPE maintenance medium (termed RPE medium)[65] containing 2% heat inactivated foetal bovine serum (FBS; Gibco). The medium was changed every 3 to 4 days and maintained at 37 °C under 5% $CO_2$. ES-RPE cells were passaged monthly, and all following experiments were conducted using passage 2 cells aged 1 to 3 months.

**Cellular internalisation studies**

*Cell uptake analysis by flow cytometry*. ES-RPE cells were seeded in 12-well plates at a density of approximately 50,000 cells/cm² in DMEM/F12 (1:1) with 3% FBS and 1% Penicillin-Streptomycin (Gibco) (termed basal medium) and incubated for 16–20 h at 37 °C under 5% $CO_2$. The basal media was then removed, and cells were washed with PBS twice followed by incubation with different concentrations of

fluorescein conjugated-poly(CEP) (F-poly(CEP), 0.001, 0.01, 0.1, 0.5, and 1 wt% dissolved in basal media). ES-RPE cells incubated with basal medium only were used as controls. After 4 h incubation at 37 °C, cells were rinsed thrice with PBS, dissociated with 0.25% (w/v) trypsin-EDTA solution (Thermo Fisher Scientific), and harvested. Cells were pelleted by centrifugation and washed in PBS once. They were then resuspended in fresh PBS containing 7-Aminoactinomycin (7-AAD) as the live/dead stain. Intracellular fluorescence intensity (mean fluorescence intensity, MFI) was determined by a FACSCelesta™ cell analyser (Becton Dickinson) at 488 nm excitation wavelength and 530 nm emission wavelength. For each sample, 10,000 events were collected, and data were analysed in quadruplicates. The gating strategy is demonstrated in Supplementary Fig. 11.

*Endocytosis inhibitor test*. ES-RPE cells were seeded and grown as per the cellular uptake analysis assay. 24 h after seeding, cells were pre-incubated at 4 °C or exposed to either Chlorpromazine (50 μM), Genistein (100 μM), Amiloride (100 μM) or Methyl-β-cyclodextrin (1 mM) (Sigma-Aldrich) for 30 min and then treated with 1 wt% F-poly(CEP) for another 4 h at 4 and 37 °C, respectively. Cells treated with media only served as control. All groups were processed as per described previously for flow cytometry.

*Internalisation analysis by immunofluorescence*. ES-RPE cells were seeded on 12-well chamber slides (Ibidi) at a density of approximately 50,000 cells/cm² and incubated for 24 h. The basal media was replaced with basal media containing 1 wt% F-poly(CEP) for another 24 h. Chloroquine (50 μM; Sigma-Aldrich) was added during the last 4 h and LysoTracker™ Red DND-99 (75 nM; Thermo Fisher Scientific) was added during the last 1 h. At the endpoint, cells were fixed with 4% paraformaldehyde (PFA; Sigma Aldrich) for 10 min and proceeded with IF.

**TNF-α and TGF-β1 (TNT)-induced model of proliferative vitreoretinopathy (PVR)**. ES-RPE cells were seeded at 50,000 cells/cm² of an uncoated 24-well culture plate (Corning) in basal media. After 16–20 h, the medium was replaced by 10 ng/mL Recombinant Human Tumour Necrosis Factor-alpha (TNF-α; R&D Systems) and 10 ng/mL Recombinant Human Transforming Growth Factor-beta1 (TGF-β1; PeproTech) in basal media and further cultured for 72 h to induce membrane formation[41]. For direct exposure, poly(CEP) (0.1, 0.5 and 1 wt%) were pre-dissolved in basal media without TNT as a polymer control or with TNT (poly(CEP) + TNT). For indirect exposure, 10 wt% poly(CEP) hydrogel was placed in a Millicell Hanging Cell Culture Insert (400 nm pore size; Merck Millipore) with TNT added into the exterior well (10 wt% poly(CEP) + TNT). The treated cells from direct exposure were placed in an incubator equipped with a time-lapse microscope (IncuCyte® ZOOM). Cells were imaged at predetermined locations within the wells every 1–2 h for a total of 72 h. At the endpoint, phase contrast images of each group were acquired with Zeiss Axio Vert.A1 microscope equipped with a digital camera and cells were then fixing with 4% PFA for 10 min. The number of membranes per well were counted by two independent blinded evaluators.

**IF of PFA-fixed ES-RPE cells**. PFA-fixed cells were blocked with 3% BSA in PBS for 1 h at RT followed by incubation with primary antibodies diluted in 0.5% BSA (Supplementary Table 4) overnight at 4 °C. Next, they were incubated with the corresponding Alexa Fluor conjugated secondary antibodies diluted in PBS (Supplementary Table 4) at RT for 30 min. Cells stained with secondary antibody only served as negative controls. Nuclei were stained with Hoechst 33342. Fluorescence images were captured at 3 to 5 random areas per well using Zeiss Axio Vert.A1 microscope equipped with a digital camera.

**Cell proliferation studies**. ES-RPE cells were seeded in triplicates at 50,000 cells/cm² in a 96-well microtiter plate (Greiner Bio-One) in basal media with 5% FBS. After 16 h, the medium was removed and replaced by each concentration (0.001, 0.01, 0.1, 0.5, and 1 wt%) of poly(CEP) dissolved in basal media with 5%

FBS. Media only served as a control. The plates were incubated at 37 °C under a time-lapse microscope (IncuCyte® ZOOM). Experiments were conducted for 72 h with data collection every 6–8 h. Using the 10× objective, three phase-contrast images were taken per time point at predetermined locations. At the endpoint, cells were fixed with 4% PFA for 10 min, followed by Ki67 IF.

**Image analysis for cell proliferation**. Two separate image processing pipelines for quantitative analysis of phase contrast (quantifying cell confluence with respect to time) and fluorescence images (quantify nuclei, Ki67 positive cell populations and quantify NRF2 nuclear intensity) were developed using Image Processing Toolbox™ in MATLAB® (MathWorks; version 2019b). The phase contrast pipeline first removed noise using Gaussian filter ($\sigma = 2$). The cellular regions had high local entropy (pixel-wise) in the 9-by-9 neighbourhood. The pipeline then segmented cellular region using entropy filter and thresholding along with morphological operations to determine the mean confluence (%) that was plotted against time (h). Similarly, fluorescence image pipeline removed noise using Gaussian filter ($\sigma = 1$). The cells here also had high local entropy (pixel-wise, 405 channel) and hence could be segmented using entropy filter and thresholding along with morphological operations to determine the mean nuclei count. The individual cells were quantified as proliferating cells (Ki67 positive) if their mean intensity in 488 channel was higher than 10. Both 405 and 488 channels were overlaid and Ki67 positive nuclei (%) was plotted. Whereas for NRF2 nuclear localisation, 405 (nuclei) and 488 (NRF2) channels were overlaid, and the nuclear MFI was calculated.

**Cell cytotoxicity assay**. ES-RPE cells were seeded and exposed to poly(CEP) solutions (0.001, 0.01, 0.1, 0.5, and 1 wt%) as described previously. Basal media only was used as control. At the endpoint of 72 h, lactate dehydrogenase (LDH) leakage into the medium, was assessed with the Cytotoxicity LDH Assay Kit (Dojindo) according to the manufacturer's protocol. In brief, absorbance was measured at 490 nm (for LDH released) and subtracted from 690 nm (for background) using multi-mode microplate reader (Biotek; Synergy HTX).

**Scratch wound healing assay**. ES-RPE cells were seeded at 300,000 cells/cm² in a 96-well Image Lock plate (Essen BioScience). Upon reaching 100% confluence, cells were scratched using the WoundMaker™ (Essen BioScience) according to the manufacturer's instructions. Thereafter, cells were washed with basal media and replaced with either basal media only (served as a control), media with poly(CEP) (0.1 or 1.0 wt%), TNT or poly(CEP) (0.1 or 1.0 wt%) + TNT. Scratch wound healing data were collected every 1–2 h for 48 h by IncuCyte® ZOOM with a scratch wound software module (Essen BioScience). Phase contrast images, which fully captured the scratch wound and surrounding cellular environment, were taken per well per time point at the same location. Data was analysed using MATLAB (MathWorks; version R2019a) using the methods of frequency filtering and mathematical morphology to approximate the boundaries of cellular regions by analysing and comparing the image data inputs. The function was modified from a prior work[66]. Wound recovery (%) was calculated using $[(A_{t=0h} - A_{t=\Delta h})/A_{t=0h}] \times 100$, where $A_{t=0h}$ was the area of the wound measured immediately after scratching (time zero), and $A_{t=\Delta h}$ was the area of the wound measured 12, 24, and 36 h after the scratch was performed.

**Reverse transcription quantitative real-time polymerase chain reaction (RT-qPCR)**

*ES-RPE cell and rabbit tissue lysis, and RNA purification*. Untreated ES-RPE cells or after TNT and/or poly(CEP) treatment were directly lysed at endpoint with 350 μL of RLT buffer (Qiagen) containing 1% β-mercaptoethanol (Sigma-Aldrich). Whereas, RPE-choroid and retina tissues were harvested from poly(CEP)-filled rabbit eyes (without PVR induction) by directly enucleated after euthanising the animals at 1 month ($n = 3$) and 2 months ($n = 3$) post-surgery. Respective tissues from non-operated eyes ($n = 3$) served as controls. Enucleated rabbit eyes were immediately dissected to remove the anterior segment containing the cornea, iris and lens. The eye cups were cut into a flower shape before gently pulling out the retinas. The RPE-choroid patches were next gently peeled from the underlying sclera. Each tissue was then lysed with 600 μL of RLT buffer containing 1% β-mercaptoethanol. Total RNA from both ES-RPE cells and rabbit tissues were extracted and purified with RNeasy Mini Kit (Qiagen) following the manufacturer's recommended protocol.

*cDNA synthesis and qPCR*. 500 ng of purified RNA from ES-RPE cells was reverse transcribed to cDNA using iScript Reverse Transcription Kit (Bio-Rad). This was followed by quantitative PCR (qPCR) with gene specific primers (Supplementary Table 5) using KAPA SYBR FAST qPCR Master Mix (2X) Kit (Sigma-Aldrich) on QuantStudio 5 Real-Time PCR system (Thermo Fisher Scientific). Experiments were performed in technical triplicates. Data analysis was performed using the comparative CT method. The expression levels were normalised to those of housekeeping gene, *Gapdh*.

**Whole-transcriptome sequencing and analysis**

*RNA sequencing (RNA-Seq)*. For total mRNA extraction from ES-RPE cells, cells were subjected to four different conditions, at time point 8 and 24 h. These

conditions include: media only, 1 wt% poly(CEP) alone, TNT alone and 1 wt% poly(CEP) + TNT. Each condition had three replicates. For total mRNA extraction from tissues, RPE-choroid and retina tissues were collected from three conditions: non-operated control rabbit eyes, and 1 month and 2 month poly(CEP)-filled eyes. Each condition had three biological replicates. Total mRNA for both cells and tissues was extracted as per the procedure in RT-qPCR above. Preliminary quality control (QC) was done with Agarose Gel Electrophoresis and sample integrity was evaluated with Agilent 2100 Bioanalyzer (minimum RIN requirement ≥8). Illumina sequencing libraries were constructed using NEBNext® Ultra™ II Directional RNA Library Prep Kit (NEB #E7765). The samples were sequenced on an Illumina NovaSeq 6000, and we obtained approximately 20 million reads per sample.

*Bioinformatics*. Sequenced reads were trimmed using Trim Galore (https://github.com/FelixKrueger/TrimGalore) and mapped to the human reference genome (GRCh38) using HISAT2[67]. The same approach was used to map reads from the in vivo RNA-Seq experiment to the rabbit reference genome (OryCun2.0). Read counts per gene were quantified using HTSeq[68] and a variance-stabilising transformation[69] was used to normalise the data. Differential expression analysis between the four different conditions was performed using DESeq2[69], and an R implementation[70] of GSEA[71] was performed on genes pre-ranked by the Wald statistic. All R scripts were run using RStudio.

**Statistical analysis**. All data are reported as mean ± s.d. Data normality were assumed for all sample sets. Statistical significance between two samples was determined using Student's unpaired *t* test. For comparison between more than two samples, one-way analysis of variance (ANOVA) followed by pairwise testing with Tukey's honest significance difference (HSD) post hoc test was done. *P*-values below 0.05 were considered significant. All analyses were performed using GraphPad Prism (ver. 8.1.1). Sample sizes for each experiment are presented in the corresponding figure caption.

**Reporting summary**. Further information on research design is available in the Nature Research Reporting Summary linked to this article.

## Data availability
The authors declare that all data supporting the results in this study are available within the paper and its Supplementary Material, or from the corresponding author upon reasonable request. Human reference genome (GRCh38) for in vitro RNA-Seq and rabbit reference genome (OryCun2.0) for in vivo RNA-Seq, and WikiPathways gene sets from "MSigDB [http://www.gsea-msigdb.org/]" were used. In vitro RNA-Seq data have been deposited to the Gene Expression Omnibus (GEO), with the dataset identifier "GSE176513". In vivo RNA-Seq data have been deposited to the GEO, with the dataset identifier "GSE200476". Source data, R codes, and RNA-seq data are provided with this paper. Source data are provided with this paper.

## Code availability
All plots were generated using R package "ggplot2" and "pheatmap". All schematics were illustrated using Adobe Illustrator CC 2015.

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

## Acknowledgements

We would like to acknowledge the veterinary team at the Translational Pre-Clinical Model Platform (Singapore Eye Research Institute, Singapore) for providing support in rabbit surgery preparation and animal follow-up. We are thankful to Central Imaging Facility at Institute of Molecular and Cell Biology (IMCB) for providing microscope resources. We also thank Xavier Le Guezennec from Frederic Bard's lab at IMCB for kindly sharing and assisting with the Incucyte® Live-Cell Analysis System. This work was supported by the National Research Foundation (NRF), Singapore, under its Competitive Research Programme (CRP) [NRF-CRP21-2018-00103] and Agency for Science, Technology and Research (A*STAR), Singapore, HMBS Domain under its Industry Alignment Fund Pre-Positioning Programme(IAF-PP) [H20/H7/a0/034] awarded to X.S.

## Author contributions

Conception and design of the study: B.H.P., Z.L., T.A.B., W.H., G.L., X.J.L., and X.S. Collection and/or assembly of data: B.H.P., Z.L., Q.L., M.S., J.Y.O., K.H.H., J.W.L., H.B., K.C.T., J.Y.C.L., K.X., A.A., S.R., K.R., D.S.L.W., Q.S.W.T., Z.Z., and G.L. Data analysis and interpretation: B.H.P., Z.L., Q.L., P.B., M.S., J.W.L., H.B., B.Y., A.D.J., V.A.B., W.Y., K.H.C., T.A.B., W.H., G.L., X.J.L., and X.S. Administrative support: V.A.B. Manuscript writing: B.H.P., Z.L., Q.L., P.B., M.S., V.A.B., K.H.C., W.H., G.L., X.J.L., and X.S. Final approval of manuscript: All authors. Funding acquisition: X.S.

## Competing interests

The authors declare no competing interests.
