## [Peer Review File · Nature Communications]

REVIEWER COMMENTS

Reviewer #1 (Remarks to the Author):

The manuscript is interesting, novel and the methods were clear. The authors did a great job. Please see the comments below;

- Introduction section: Explain the role of NRF-2 in the introduction. How its activation affects the retinal scarring and its role in PVR. It helps the reader to connect the drug delivery aspect (biofunctional polymer) with molecular biology.
- Add time in the synthetic scheme (Supplement S1 figure)
- Provide the XRD of the poly (CEP). This would help to understand the properties of the polymer and the behavior of degradation.
- Did the author analyses the degradation product of the poly (CEP)? Or shed Micelles in the Media
- Line 141: Are Micelles degraded by surface erosion, bulk erosion, or degradation by diffusion may be possible?
- Line 279: 'in' repeated two times in a sentence.
- Not able to read figure 8

Reviewer #2 (Remarks to the Author):

PVR is an important problem in ophthalmology. This work is an important contribution to the problem. I have few rather important but easily addressable comment:
It is imperative that negative comments on surgery in PVR will be removed or tone down. It is the only hope for many and disparaging the current procedure is unnecessary. Many exp are not quantifiable (imaging can be presented only once) and throughout the manuscript n number of animals needs to be described and commented on similarity of results. Toxicity (toxicology) of the polymer must be addressed. The characterization of the polymer and chemistry should be included as a separate figure even if it was published previously in some of the most critical aspects.

Reviewer #3 (Remarks to the Author):

This is an excellent study on the ability of the biofunctional polymer, poly(ethylene glycol) (PEG), poly(propylene glycol) (PPG) and poly(ϵ -caprolactone) (PCL), termed poly(CEP). The authors have utilized in vivo experiments to show poly(CEP) inhibits PVR formation in a rabbit model of PVR. The poly CEP treated eyes did not develop PVR compared to SF6 and Air filled eyes based on representative photos, OCT, and histology. In vitro experiments were performed to demonstrate the ability of poly(CEP) to inhibit various aspects of PVR pathogenesis and study the mechanism of action.

- 1) It would be helpful to have grading of PVR based on published grading schemes for rabbit PVR to compare the in vivo groups quantitatively
- 2) The in vitro studies suggest a NRF2 dependent mechanism - can the in vivo sections be analyzed to assess if this is occurring in vivo
- 3) The authors state that "this further corroborates our hypothesis that the phenomenon of in vitro PVR reversal is mediated primarily by poly(CEP) induced gene expression." It is not clear to me that the experiment as structured shows reversal vs. prevention of gene expression. Furthermore, gene expression alone would not necessarily support reversal. It would be helpful to assess functional changes.

Reviewer comments

Reviewer #1 (Remarks to the Author):

The manuscript is interesting, novel and the methods were clear. The authors did a great job. Please see the comments below;

We would like to thank Reviewer #1 for the supportive comments on the novelty and approach of this work.

- Introduction section: Explain the role of NRF-2 in the introduction. How its activation affects the retinal scarring and its role in PVR. It helps the reader to connect the drug delivery aspect (biofunctional polymer) with molecular biology.

We thank Reviewer #1 for this suggestion. NRF2 has only been previously described to be involved in general EMT, but not PVR. Indeed, the involvement of NRF2 in PVR is a novel finding highlighted by our manuscript. Nonetheless, we have modified our introduction to include a brief introduction of NRF2 role in general EMT, and included **line 65 to 72 (Introduction, Pg 4)** to clarify this.

- Add time in the synthetic scheme (Supplement S1 figure)

We thank Reviewer #1 for this suggestion. The reaction duration has been included into the synthetic scheme in **Supplementary Fig. 1a**.

- Provide the XRD of the poly (CEP). This would help to understand the properties of the polymer and the behavior of degradation.

We thank Reviewer #1 for this suggestion. As our poly(CEP) micelles are expected to be arranged in an amorphous state, it is unlikely that the XRD will demonstrate any crystal peaks. In fact, XRD has been performed by other groups on polymeric micelles similar to poly(CEP) and have consistently demonstrated absence of crystalline regions in their polymeric micelles (PMID: 21283869; 29978290; 25980982). Our group has previously performed small angle X-ray scattering (SAXS) on poly(CEP) micelles. The results suggest that the amphiphilic copolymers first form spherical micelles above the critical micelle temperature (CMT) and these micelles further aggregate into rod-like micelles at temperatures above gelation. A statement describing this potential behaviour of poly(CEP) micelles has been added to **line 163 to 165 (Results, Pg 8)** and **line 380 to 382 (Discussion, Pg 18)**.

- Did the author analyses the degradation product of the poly (CEP)? Or shed Micelles in the Media

We thank Reviewer #1 for this comment. In this manuscript, we are analysing the shed micelles in the media (**Supplementary Fig. 5b – d**). Analysis of the degradation products of poly(CEP) has been addressed in our previous publications. We have further clarified this in **line 153 to 155 (Results, Pg 8)**, stating that poly(CEP) fully degrades into its oligomeric form (PMID: 34810039).

- Line 141: Are Micelles degraded by surface erosion, bulk erosion, or degradation by diffusion may be possible?

We thank Reviewer #1 for this comment. The main mode of poly(CEP) degradation is likely to be surface erosion. Poly(CEP) is a supramolecular hydrogel, of which the polymeric matrix is held together by non-covalent bonds. Degradation occurs primarily via surface erosion, secondary to dilution effects on the gel surface. In addition, hydrolysis of the ester bonds contained within the PCL segments is likely to contribute to poly(CEP) degradation. However, this latter process has been shown to be slow and often requires the presence of enzymes to accelerate its biodegradation (PMID: 31843602; PMID: 28623631). Thus, surface erosion remains the primary mode of degradation for poly(CEP). We have clarified this in **line 152 to 155 (Results, Pg 8)**.

- Line 279: 'in' repeated two times in a sentence.

We thank Reviewer #1 for this comment. We have removed the repetition.

- Not able to read figure 8

We thank Reviewer #1 for this comment. We have increased the font size for text in Figure 8 (Supplementary).

Reviewer #2 (Remarks to the Author):

PVR is an important problem in ophthalmology. This work is an important contribution to the problem. I have few rather important but easily addressable comments:

We thank Reviewer #2 for recognising the importance of our work towards the understanding and treatment of PVR.

- It is imperative that negative comments on surgery in PVR will be removed or tone down. It is the only hope for many and disparaging the current procedure is unnecessary.

We thank Reviewer #2 for this comment. We have toned down our comments on surgery in PVR in **line 43/44 (Abstract, Pg 3)** and **line 63/64 (Introduction, Pg 4)**.

- Many experiments are not quantifiable (imaging can be presented only once) and throughout the manuscript a number of animals needs to be described and commented on similarity of results.

We are grateful to Reviewer #2 for this comment. For the *in vivo* results presented in **Fig. 1**, we have included “*n*” numbers within the manuscript (**Results – line 99 to 148**). In addition, we have revised **Supplementary Table 2**, to include surgical findings and PVR grading for all operated rabbits, demonstrating similarity and consistency within each surgical group.

- Toxicity (toxicology) of the polymer must be addressed. The characterization of the polymer and chemistry should be included as a separate figure even if it was published previously in some of the most critical aspects.

We are grateful to Reviewer #2 for the comment. Poly(CEP) toxicity in the rabbit eye has been addressed in **Supplementary Fig. 3**, whereby we demonstrated biocompatibility for up to 2 months, using *in vivo* ophthalmic imaging and histology (refer to **line 123 to 125, Results, Pg 7**). Briefly, rabbits implanted with Poly(CEP) demonstrated normal retina on both colour fundus and IR imaging. In addition, retinal structure was well-maintained confirmed by H&E.

Characterisation of poly(CEP) and its chemistry has been included in **Supplementary Fig. 1, 4 and 5**.

Reviewer #3 (Remarks to the Author):

This is an excellent study on the ability of the biofunctional polymer, poly(ethylene glycol) (PEG), poly(propylene glycol) (PPG) and poly(ϵ -caprolactone) (PCL), termed poly(CEP). The authors have utilized *in vivo* experiments to show poly(CEP) inhibits PVR formation in a rabbit model of PVR. The poly CEP treated eyes did not develop PVR compared to SF6 and Air filled eyes based on representative photos, OCT, and histology. *In vitro* experiments were performed to demonstrate the ability of poly(CEP) to inhibit various aspects of PVR pathogenesis and study the mechanism of action.

We thank Reviewer #3 for affirming the importance of our work in describing the role of poly(CEP) to prevent PVR.

- It would be helpful to have grading of PVR based on published grading schemes for rabbit PVR to compare the *in vivo* groups quantitatively

We are grateful to Reviewer #3 for the comment. We have performed PVR grading for all rabbits according to published grading schemes and summarised this in **Supplementary Table 2** (refer to **Results, Pg 6, line 106 to 107**).

- The *in vitro* studies suggest a NRF2 dependent mechanism - can the *in vivo* sections be analyzed to assess if this is occurring *in vivo*

We are grateful to Reviewer #3 for this suggestion. To provide additional evidence of poly(CEP)-induced NRF2 dependent mechanism *in vivo*, we have performed a comprehensive RNA-Seq analysis detailing the upregulation of NRF2 target genes in the rabbit tissue samples of retina and RPE exposed to poly(CEP) *in vivo*. This data is included in **Supplementary Fig. 10** and described in **line 314 to 319 (Results, Pg 14)**.

- The authors state that "this further corroborates our hypothesis that the phenomenon of *in vitro* PVR reversal is mediated primarily by poly(CEP) induced gene expression." It is not clear to me that the experiment as structured shows reversal vs. prevention of gene expression. Furthermore,

gene expression alone would not necessarily support reversal. It would be helpful to assess functional changes.

We thank Reviewer #3 for the comment. Our experimental design, as mentioned in **line 507 to 509 (Methods, Pg 23)**, involved exposing RPE cells to both TNT and poly(CEP) simultaneously. Therefore, it only tests the prevention of gene expression induced by TNT, and not reversal. Thus, as suggested by Reviewer #3, we have replaced the word 'reversal' with 'prevention' in that statement in **line 291 (Results, Pg 13)**.

REVIEWERS' COMMENTS

Reviewer #1 (Remarks to the Author):

The authors have responded to the reviewer's comments satisfactorily. Great team work!

Reviewer #2 (Remarks to the Author):

The authors made appropriate changes. It is an important paper. It was pleasure reviewing it.

Reviewer #3 (Remarks to the Author):

The authors have appropriately responded to my previous concerns.